



# Cereal-legume mixtures increase net CO$_2$ uptake in a forage system of the Eastern Pyrenees

**Mercedes Ibañez[1, 2]\***; **Núria Altimir[2a]**; **Àngela Ribas[2,3,4]**; **Werner Eugster[5]**; **Maria-Teresa Sebastià[1,2]**

[1]GAMES group, Dept. HBJ, ETSEA, University of Lleida (UdL). Av. Alcalde Rovira Roure, 191, 25198, Lleida, Spain.

[2]Laboratory of Functional Ecology and Global Change (ECOFUN), Forest Science and Technology Centre of Catalonia (CTFC). C/ de Sant Llorenç, 0, 25280 Solsona, Lleida, Spain.

[3]Universitat Autònoma de Barcelona, 08193, Bellaterra, Spain.

[4]Centre for Ecological Research and Forestry Applications (CREAF), 08193, Bellaterra, Spain.

[5]ETH Zürich, Institute of Agricultural Sciences, Universitätstrasse 2, 8092, Zürich, Switzerland.

[a]Current address: Institute for Atmospheric and Earth System Research (INAR), University of Helsinki, Physicum, Kumpula campus, Gustaf Hällströmin katu, 2 , 00560, Helsinki, Finland.

\*Corresponding author; e-mail: mercedes.ibanez@ctfc.es



**Abstract**
Forage systems are the major land use, and provide essential resources for animal feeding. Assessing the
influence of forage species on net ecosystem $CO_2$ exchange (NEE) is key to develop management
strategies that can help to mitigate climate change, while optimizing productivity of these systems.
However, little is known about the effect of forage species on $CO_2$ exchange fluxes and net biome
production (NBP), considering: species ecophysiological responses; growth and fallow periods
separately; and the management associated with the particular sown species. Our study assesses the
influence of cereal monocultures *vs.* cereal-legume mixtures on (1) ecosystem-scale $CO_2$ fluxes, for the
whole crop season and separately for the two periods of growth and fallow; (2) potential sensitivities of
$CO_2$ exchange related to short-term variations in light, temperature and soil water content; and (3) NBP
during the growth period; this being the first long term (seven years) ecosystem-scale $CO_2$ fluxes dataset
of an intensively managed forage system in the Pyrenees region. Our results provide strong evidence that
cereal-legume mixtures lead to higher net $CO_2$ uptake than cereal monocultures, as a result of higher gross
$CO_2$ uptake, while respiratory fluxes did not significantly increase. Also, management associated with
cereal-legume mixtures favoured vegetation voluntary regrowth during the fallow period, which was
decisive for the cumulative net $CO_2$ uptake of the entire crop season. All cereal-legume mixtures and
some cereal monocultures had a negative NBP (net gain of C) during the growth period, indicating C
input to the system, besides the yield. Overall, cereal-legume mixtures enhanced net $CO_2$ sink capacity of
the forage system, while ensuring productivity and forage quality.

**Key words:** ecosystem respiration ($R_{eco}$), gross primary production (GPP), light response, management,
monocultures, net ecosystem $CO_2$ exchange (NEE).



## 1. Introduction

Forage systems, including feed crops together with intensively and extensively managed pasturelands, are the major land use, covering about 30% of the world's terrestrial surface and 80% of agricultural land (Steinfeld and Wassenaar, 2007). Thus, assessing the role of forage species on the carbon (C) balance of these systems is essential to develop management strategies that can mitigate climate change, while optimizing productivity. To this regard, forage mixtures have been generally associated with higher productivity than monocultures (Brophy et al., 2017; Finn et al., 2013; Kirwan et al., 2007; Ribas et al., 2015), resulting from higher resource use efficiency, including light (Hofer et al., 2017; Milcu et al., 2014), water (Chapagain and Riseman, 2015; Liu et al., 2016), and nitrogen (Sturludóttir et al., 2013; Suter et al., 2015). Mixtures have also been described to present lower rates of weed invasion (Connolly et al., 2018; Frankow-Lindberg et al., 2009; Kirwan et al., 2007). However, the role of forage species in the net ecosystem $CO_2$ exchange (NEE), as well as on NEE components — gross primary production (GPP) and ecosystem respiration ($R_{eco}$) — is less understood.

In addition, the interaction between local conditions and management practices result in high $CO_2$ exchange variability (Moors et al., 2010; Oertel et al., 2016). And, while information on the $CO_2$ budget of grasslands (Berninger et al., 2015; Imer et al., 2013; Schaufler et al., 2010) and forage crops (Ceschia et al., 2010; Kutsch et al., 2010; Vuichard et al., 2016) of central and northern Europe is rather abundant, such information is very scarce in the Mediterranean basin, even though it is a highly vulnerable region to climate change (FAO, 2010). Indeed, forage productivity in Mediterranean areas is among the lowest in Europe (Smit et al., 2008), due to important water constraints (Porqueddu et al., 2016), and more information is needed to establish management practices that may enhance C sequestration while ensuring productivity.

In addition, it is also crucial to understand the role of forage species in net biome production (NBP), accounting for all C inputs and exports ($NBP = NEE – C_{input} + C_{export}$), to assess the final C budget, beyond the NEE. In fact, many grasslands and forage crops may be acting as net $CO_2$ sinks when only assessing NEE, but they become net $CO_2$ sources when accounting for the oxidation (via digestion by animals) of total exported biomass (Ceschia et al., 2010; Kutsch et al., 2010; Moors et al., 2010).

Our study presents in this regard the first long-term (seven years) dataset of ecosystem-scale $CO_2$ fluxes of an intensively managed forage system in the Pyrenees, which combines a crop rotation of cereal species grown in monocultures and cereal-legume mixtures, with direct grazing after the harvest (fallow period). Such practices have been traditionally conducted in Mediterranean mountain regions (Sebastià et al., 2011) to increase productivity and preserve soil fertility (Sánchez et al., 2013).

Thus, our objective is to assess differences between cereals grown in monoculture and cereal-legume mixtures in (1) ecosystem-scale $CO_2$ fluxes, for the whole crop season and separately for the two periods of growth and fallow; (2) potential sensitivities of $CO_2$ exchange related to short-term variations in light, temperature and soil water content; and (3) NBP during the growth period. Also, we hypothesize that cereal-legume mixtures in comparison to cereal monocultures: (1) will show more net $CO_2$ uptake (more negative NEE); (2) this increase in the net uptake will be due to increased GPP in combination with unchanged $R_{eco}$; and (3) will show more negative NBP.

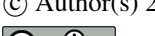



## 2. Material and methods

### 2.1 Study site and experimental design

The study site is a forage system located in the montane elevation belt of the Eastern Pyrenees, in Pla de Riart (42° 03' 48" N, 1° 30' 48" E), at 1003 m a. s. l. Climate is sub-Mediterranean (Peel et al., 2007), typical in mountain areas with Mediterranean influences, with a mean annual precipitation of 750 mm and mean annual temperature of 11 ℃ (Ninyerola et al., 2000), including the summer drought period. The soil is a petrocalcic calcixerept (Badía-Villas and del Moral, 2016).

All management events, including fertilizing, sowing and harvesting (Table 1) were reported by the manager of the site and validated by in situ visits. The site was managed by a rotation of cereals grown in monoculture and cereal-legume mixtures. Every year the yield was harvested, and during the fallow (from harvest to next sowing), the voluntary regrowth of the vegetation was extensively grazed by around 30 cattle ($\approx$ 0.91 livestock units (LSU) ha$^{-1}$) from late August to late October (Fig. 1).

Yield was estimated (Table 1) considering the productivity reported by the manager and in situ samplings after oven drying plant material at 60 ℃ until constant weight. Plant material was analysed to determine C content and forage quality indicators (Table S1). Analyses were performed by the Department of Animal and Food Science, Autonomous University of Barcelona according to standard methods (Table S1). Afterwards, C exported through yield (Table 1) was estimated, considering the yield, species proportions (Fig. 1), and species C content (Table S1). C exported through yield was used to account for the NBP (Sect. 2.5).

### 2.2 Eddy covariance measurements

The site is equipped with an eddy covariance flux station, running since August 2010, and our study period included data from sowing of the first studied season (barley, sown 01/11/2010) until the end of the fallow period of the last studied season (oat and vetch mixture, 01/11/2017, Fig. 1). The eddy covariance flux station continuously measured the concentration of $CO_2$ (mmol m$^{-3}$) and $H_2O$ (mmol m$^{-3}$) using an open path $CO_2$ and $H_2O$ gas analyser (LI-7500, LI-COR Inc., Lincoln, NE, USA), and turbulent flux components, including wind direction and speed using a 3D sonic anemometer (CSAT-3, Campbell Scientific Inc, Logan, UT, USA) to calculate $CO_2$, $H_2O$, and energy exchange at the ecosystem level.

In addition, the station recorded ancillary meteorological variables, including incoming and outgoing shortwave and longwave radiation (NR01, Hukseflux, Delft, the Netherlands); air temperature ($T_a$, CS215, Campbell Scientific Inc, Logan, UT, USA); average soil temperature 1-20 cm ($T_s$, TCAV, Campbell Scientific Inc, Logan, UT, USA); volumetric soil water content (SWC, CS616, Campbell Scientific Inc, Logan, UT, USA); photosynthetically active radiation (PAR, SKP215, Skye Instruments Ltd, Powys, UK); and normalized difference vegetation index, calculated as NDVI = (NIR − Red) / (NIR + Red), where "Red" and "NIR" are the spectral reflectance measurements acquired in the red and near infrared regions, respectively.

Raw data provided by the sensors were processed and $CO_2$ fluxes were calculated at 30-minute averages using the EddyPro software (LI-COR Inc, Lincoln, NE, USA). Negative values refer to the flux from the



atmosphere to the biosphere and positive values correspond to the flux from the biosphere to the
atmosphere (micrometeorological sign convention).
We applied frequency response corrections (Moncrieff et al., 2004, 1997), density fluctuation corrections
(Webb et al., 1980), and determination of data quality using the Foken et al., (2004) approach. The
Foken et al. (2004) approach suggests a quality scale ranging from 1 (highest data quality) to 9 (poorest
data quality), and records with quality 7 or higher were excluded (Papale, 2012). Also, $CO_2$ fluxes outside
a physically realistic range ($\pm$ 50 µmol m$^{-2}$ s$^{-1}$) were rejected.
We inspected night-time (PAR < 5 µmol photons m$^{-2}$ s$^{-1}$) $CO_2$ fluxes, as they tend to be underestimated
under low turbulence (Aubinet et al., 2012), conditions that can be frequent at night. We carefully
examined the possibility of a low turbulence effect assessing the existence of an $u_*$ threshold at all
recorded $T_s$ classes (Reichstein et al., 2005), ranging from –3 to 34 ℃ in 1 ℃ intervals. Relevant $u_*$
thresholds were not detected. In addition, we inspected night-time $CO_2$ fluxes in order to detect possible
outliers and calculated the 0.025, 0.25, 0.5, 0.75 and 0.975 quantiles for each $T_s$ class. Data below the
lowest (0.025) or the highest (0.975) quantile were excluded from further analysis.
Data were filtered according to the footprint, based on the Kljun model (Kljun et al., 2004), including all
the fluxes in which more than 80% of the contribution came from the study field (Göckede et al., 2008).
After all data cleaning and filtering, retained data for further analysis were a 65% of all the available data,
ranging between 81% and 53% depending on year (Table S2).
Afterwards, we gap-filled NEE data using the sMDSGapFill function (Reichstein et al., 2005) of the
REddyProc package (Wutzler et al., 2018) for R software (R core Team, 2017). The goodness of the
gap-filling was also inspected comparing observed NEE data with their theoretically predicted data by
gap-filling (see an example in Fig. S1). Gap-filled NEE data were also partitioned into GPP and $R_{eco}$,
using the night-time based partitioning approximation, SMRFLuxPartition equation, also of the
REddyProc package.
In line with our first objective, we described NEE, GPP and $R_{eco}$ dynamics, and performed budgets
(expressed in g C m$^{-2}$) for each: (a) crop season — from sowing to sowing —, (b) growth period — from
sowing to harvesting —, and (c) fallow period — from harvesting to sowing. Note that in 2014 systematic
data gaps occurred due to energy supply problems, for which NEE, GPP and $R_{eco}$ budgets could not be
calculated. However, 2014 gap-filled data were used to describe $CO_2$ exchange dynamics, and 2014 real
recorded data were included in all the modelling.
**2.3 Net ecosystem $CO_2$ exchange modelling: diversity-interaction model**
Species can drive ecosystem functions via species identity effects, but also via species interactions and
complementarity effects (Kirwan et al., 2007; Orwin et al., 2014; Wolfgang et al., 2017). Thus, also in
line with our first objective we disaggregated the influence of cereal monocultures form cereal-legume
mixtures on NEE using a diversity-interaction approach (Kirwan et al., 2007, 2009). The approach
compares a null model, in which a change in the diversity has no effect on the response variable, with
models that address the diversity influence at different levels.
In our study we compared the null model Eq. (1), in which NEE (µmol $CO_2$ m$^{-2}$ s$^{-1}$) depended only on
environmental variables, including $T_a$ (℃), net radiation ($R_{net}$, W m$^{-2}$), SWC (fraction), vapour pressure





deficit (VPD, hPa), and time — considering time as crop season — with a diversity-interaction model,
which included species identity and species interaction effects Eq. (2).

$$NEE = \beta_{T_a}T_a + \beta_{R_{net}}R_{net} + \beta_{SWC}SWC + \beta_{VPD}VPD + \beta_{time}time + \varepsilon$$

(Equation 1. Null model)

$$NEE = Null\ model + \beta_B P_B + \beta_T P_T + \beta_W P_W + \beta_{OV}P_{OV} + \beta_{TOV}P_{TOV} + \varepsilon$$

(Equation 2. Diversity-interaction model)

Here $P$ indicates species proportions and the sub-index $B$ indicates barley, $T$ triticale, $W$ wheat, $O$ oat and
$V$ vetch respectively. The models were run without intercept in order to test the effect of all the species
proportions at the same time.
A preliminary modelling showed that SWC and time could be excluded from the null model Eq. (1), since
the inclusion of these variables did not provide a better fitting. Then, the null model Eq. (1) and the
diversity-interaction model Eq. (2) were compared by an analysis of variance (ANOVA) to account for
the most parsimonious and explanatory model. The diversity-interaction model was significantly different
from the null model (F = 7.65, p < 0.001); therefore, the final model was the diversity-interaction model,
which included the proportion of each forage species and its interactions, in addition to environmental
variables ($T_a$, $R_{net}$, VPD).
The approach was run on all observed data (30-minute average); on daily-averaged data; and on
weekly-averaged data. The model performed the best fitting (best adjusted $R^2$) when using
weekly-averaged data, probably due to a considerable day-to-day variability of the environmental
variables and $CO_2$ fluxes. Also, considering that the main goal of this analysis was to assess the influence
of forage species on NEE, whose influence is probably more noticeable at a seasonal scale, we present the
model run on the weekly-averaged data, as it was able to reduce noise and extract the influence of forage
species with greater reliability.
**2.4 $CO_2$ exchange response to light, temperature and soil water content**
In line with our second objective, we explored differences between cereal monocultures and
cereal-legume mixtures from a mechanistic perspective, modelling separately light response of observed
$CO_2$ fluxes during daytime (termed as $NEE_{day}$ in what follows), and $T_s$ and SWC response of night time
fluxes (termed as $R_{eco,night}$ in what follows) as explained below.

**Biogeosciences** Open Access
Discussions
EGU

### 2.4.1 $NEE_{day}$ light response

$NEE_{day}$ (PAR > 5 µmol photons $m^{-2}$ $s^{-1}$) light response was modelled using a logistic sigmoid response function (Moffat, 2012), which models $NEE_{day}$ (µmol $CO_2$ $m^{-2}$ $s^{-1}$) as function of PAR Eq. (3).

$$NEE_{day} = -2 \cdot GPP_{sat} \cdot \left( -0.5 + \frac{1}{1 + e^{\frac{-2 \cdot \alpha \cdot PAR}{GPP_{sat}}}} \right) + R_{eco,day}$$

(Equation 3)

Here $GPP_{sat}$ (µmol $CO_2$ $m^{-2}$ $s^{-1}$) is the asymptotic gross primary production, $\alpha$ (dimensionless) is the apparent initial quantum yield, defined as the initial slope of the light-response curve, and $R_{eco,day}$ (µmol $CO_2$ $m^{-2}$ $s^{-1}$) the average daytime ecosystem respiration. Light response parameters ($GPP_{sat}$, $\alpha$ and $R_{eco,day}$) were calculated for each day and crop season, using the nlsList function of the nlme package (Pinheiro et al., 2015). Parameters whose estimates were not significantly different from zero ($p \geq 0.05$) were discarded from further analysis.

Afterwards, we described light response dynamics and assessed differences on the light response parameters between cereal monocultures and cereal-legume mixtures for each period (growth and fallow). For that purpose we ran an ANOVAs and tukey post-hoc tests, using the HSD.test function of the agricolae package (Mendiburu, 2017), with the given parameter ($GPP_{sat}$, $\alpha$ and $R_{eco,day}$) as a function of forage type (cereal monoculture and cereal-legume mixture) in interaction with period (growth and fallow).

### 2.4.2 $R_{eco,night}$ response to temperature and soil water content

A preliminary overview of $R_{eco,night}$ (PAR < 5 µmol photons $m^{-2}$ $s^{-1}$) suggested that $R_{eco,night}$ increased with $T_s$ at $T_s < 20°C$, but decreased above this threshold. Therefore, we modelled $R_{eco,night}$ (µmol $CO_2$ $m^{-2}$ $s^{-1}$) as a function of $T_s$ (°C) and SWC (fraction) using the equations proposed by Reichstein et al. (2002), which consider changes in the temperature sensitivity depending on soil moisture Eq. (4-6).

$$R_{eco,night} = R_{eco,ref} \cdot f(T_s, SWC) \cdot g(SWC)$$

(Equation 4)

$$f(T_s, SWC) = e^{E_0(SWC) \cdot \left( \frac{1}{T_{ref} - T_0} - \frac{1}{T_s - T_0} \right)}$$

(Equation 5)

$$g(SWC) = \frac{SWC - SWC_0}{(SWC_{1/2} - SWC_0) + (SWC - SWC_0)}$$

(Equation 6)

Here the activation energy, $E_0$ (°$C^{-1}$), is a linear function of SWC ($E_0 = a+b \cdot SWC$); $T_{ref}$ is the reference temperature, set as the mean temperature of a given period, here set as the mean $T_s$ of the entire





measuring period ($T_{ref}$ = 12.12 °C); $T_0$ the lower limit for $R_{eco,night}$, here set at –46.02 °C, as in the original
model by Lloyd and Taylor (1994); $SWC_0$ (fraction) the soil water content below which $R_{eco,night}$ ceases;
$SWC_{1/2}$ (fraction) the soil water content at which maximal $R_{eco,night}$ halves; and $R_{eco,ref}$ (µmol $CO_2$ m$^{-2}$ s$^{-1}$)
the reference ecosystem respiration at standard conditions ($T_{ref}$) and non-limiting SWC (Reichstein et al.,
2002). $R_{eco,night}$ response parameters ($R_{eco,ref}$, $E_0$, $SWC_0$ , $SWC_{1/2}$) were calculated considering all seasons
together (2011-2017) and for each crop season, using the nlsList function.
Similarly as in the diversity-interaction model (Sect. 2.3), we performed the $R_{eco,night}$ modelling on all
observed data (30-minute average), on daily-averaged data and on weekly-averaged data. Afterwards, we
calculated $R^2$ as the linear relationship between modelled and measured observations. The model
performed best (highest $R^2$) when using weekly-averaged data, probably due to the high day-to-day
variability of $R_{eco,night}$ and $T_s$.

### 2.5 Net biome production (NBP)

Finally, in line with our third objective, we estimated the NBP during the growth period. NBP can be
estimated knowing the NEE; C exports, including harvest/grazing and other gas emissions such as
methane or volatile organic compounds; and C imports, including organic C fertilizers and sowing. In our
study, C exports through methane were expected not to be very significant, because methane effluxes
require water saturated soils, typically with standing water (Oertel et al., 2016), which was never the case;
and volatile organic compounds were expected to be negligible (Soussana et al., 2010). C inputs through
sowing and fertilizers (mostly inorganic nitrogen fertilizers, Table 1) could also be neglected as they only
represent a very small C amount.  Thus, we estimated the NBP during the growth period as the sum of the
NEE budget of that period and C exported through the yield Eq. (7).

$$NBP = NEE + Yield$$

249                                                (Equation 7)


### 3.  Results

### 3.1 Forage species influence on $CO_2$ exchange dynamics and budgets

Seasonal $CO_2$ flux dynamics evolved according to environmental conditions, forage growth and
management events (Fig. 2). Maximum net $CO_2$ uptake was achieved during spring, when temperatures
were mild, SWC increased, and the forage development reached its peak biomass (Fig. 2). $CO_2$ exchange
capacity of the system decreased with harvesting (Fig. 2.a), also showed by the drastic decrease of the
NDVI (Fig. 2.d).
The field acted as a net $CO_2$ sink throughout all the studied crop seasons (negative NEE, Fig. 3.a). NEE of
cereal-legume mixtures was more negative and less variable (−363 g C m$^{-2}$, year 2013, and
−383 g C m$^{-2}$ year 2017, Fig. 3.a) than that of cereal monocultures (ranging from −70 to −226 g C m$^{-2}$,
Fig. 3.a).
During the growth period, cereal-legume mixtures showed the highest net $CO_2$ uptake, with a NEE of –359
and −429 g C m$^{-2}$ in 2013 and 2017 respectively (Fig. 3.b). On the other hand, cereal monocultures had a
NEE that ranged from –128 to −348 g C m$^{-2}$ (Fig. 3.b), with triticale being the cereal monoculture with the
highest net uptake (−348 g C m$^{-2}$, Fig. 3.b).
During the fallow period $R_{eco}$ was the dominant flux in all cases (Fig. 3.c), although there were some
differences in the $CO_2$ exchange dynamics between cereal monocultures and cereal-legume mixtures
(Fig. 2.a), which were decisive for the cumulative net $CO_2$ uptake of the whole crop season. During the
fallow of grass-legume mixtures there was a more marked voluntary regrowth of the vegetation (Fig. 2.d)
that promoted a period of net $CO_2$ uptake after the harvest, especially strong in the triticale, oat and vetch
mixture (year 2013), and the oat and vetch mixture (year 2014, Fig. 2.a). Note that although gap-filled
2014 data were not used to account for $CO_2$ exchange budgets (Fig. 3) due to systematic gaps; 2014
gap-filled data could be used to describe $CO_2$ exchange dynamics and allowed us to identify this rebound
in the net $CO_2$ uptake during the fallow period of that year.
On the contrary, cereal monocultures generally did not show this voluntary regrowth during the fallow
period (Fig. 2.d), and gross and net $CO_2$ uptake capacity of the system decreased drastically (Fig. 2.a). The
exception was the wheat monoculture in 2015, when there was vegetation voluntary regrowth after the
harvest that resulted in net $CO_2$ uptake during the fallow period.
The diversity-interaction model (Table 2) confirmed the influence of forage species on NEE. The model
estimates indicated less net $CO_2$ uptake in cereal monocultures than in cereal-legume mixtures (Table 2,
negative sign in the estimate means uptake), again with a high variability within cereal monocultures.
Barley was the cereal monoculture with the lowest net uptake ($-1.0 \pm 0.3$ µmol $CO_2$ m$^{-2}$ s$^{-1}$, t = −3.39,
p < 0.001, Table 2) and triticale was the cereal monoculture with the highest net uptake among the
monocultures ($-1.6 \pm 0.4$ µmol $CO_2$ m$^{-2}$ s$^{-1}$, t = −4.40, p < 0.001, Table 2). Cereal-legume mixtures,
however, showed higher net $CO_2$ uptake rates (oat x vetch $-2.0 \pm 0.3$ µmol $CO_2$ m$^{-2}$ s$^{-1}$, t = −7.44,
p < 0.001, Table 2) than all cereal species in monoculture. The addition of triticale in the mixture did not
have a significant effect on NEE (Table 2).
**3.2 Cereal monocultures vs. cereal-legume mixtures: $NEE_{day}$ light response**
All three light response parameters exhibited pronounced seasonality, as result of phenological changes
and management events (Fig. 4). During the growth period, cereal-legume mixtures exhibited on average
slightly higher values of $GPP_{sat}$ than cereal monocultures, while $R_{eco,day}$ did not increase (Fig. 5).
During the fallow period, cereal-legume mixtures presented on average significantly higher $GPP_{sat}$ and
α values than cereal monocultures (Fig. 5), due to the voluntary regrowth of the vegetation (Fig. 2.d),
which also caused a rebound on $GPP_{sat}$ and α (Fig. 5).
**3.3 Cereal monocultures *vs.* cereal-legume mixtures: $R_{eco,night}$ response to temperature and soil**
**water content**
$R_{eco,night}$ models, based on the equations proposed by Reichstein et al. (2002, our Eq. 4- 6), presented a
satisfactory fitting, with $R^2$ ranging from 0.19 to 0.75 across seasons (Table 3). When assessing all seasons
together, $T_s$ and SWC drove $R_{eco,night}$ (Fig. 6); with an activation energy ($E_0$) significantly dependent on



SWC ($E_0 \sim a + b \cdot SWC$, $a = 76 \pm 40$ and $b = 483 \pm 259$ °C$^{-1}$, Table 3), indicating that temperature sensitivity
was dependent on SWC Eq. (5). Also, soil water content at which maximal $R_{eco,night}$ halves ($SWC_{1/2}$) was
significant ($0.06 \pm 0.01$, Table 3), indicating that $R_{eco,night}$ decreased to half-maximum or lower at
SWC $\leq 6 \pm 1$ %.
However, some estimates of the $R_{eco,night}$ response parameters were not significantly different from zero
($p \geq 0.05$, see significant estimates in bold, Table 3); and when assessing differences between forage
types, non-significant estimates were not considered for comparison. Yet, $E_0$ of barley, in year 2011
($b = 3668 \pm 1645$ °C$^{-1}$, Table 3), and of wheat, in year 2015 ($b = 850 \pm 627$ °C$^{-1}$, Table 3), were
significantly dependent on SWC, both values being much higher than the average of all crop seasons
($b = 483 \pm 259$ °C$^{-1}$, Table 3). Also, the reference ecosystem respiration ($R_{eco,ref}$) of triticale in year 2012,
was significantly different from zero ($4 \pm 2$ μmol $CO_2$ m$^{-2}$ s$^{-1}$, Table 3), exceeding $R_{eco,ref}$ of all seasons
together ($2.8 \pm 0.3$ μmol $CO_2$ m$^{-2}$ s$^{-1}$, Table 3). Finally, soil water content below which $R_{eco,night}$ ceases
($SWC_0$) and $SWC_{1/2}$ had a significant influence on $R_{eco,night}$ in the triticale, oat and vetch mixture
(year 2013), the oat and vetch mixture (year 2014), and in the wheat monoculture (year 2015, Table 3).
Both cereal-legume mixtures (year 2013 and 2014), had a $SWC_{1/2}$ that was very close to $SWC_0$, indicating
that SWC could reach very low values before $R_{eco,ref}$ halved, although this SWC value was already very
close to the limit at which $R_{eco,ref}$ ceases ($SWC_0$). On the contrary, during the wheat monoculture of 2015,
$SWC_{1/2}$ ($0.08 \pm 0.03$, Table 3) doubled $SWC_0$ ($0.04 \pm 0.03$, Table 3).
**3.4   Cereal monocultures *vs*. cereal-legume mixtures: Net biome production (NBP)**
Finally, NBP during the growth period indicated net C input into the system (negative NBP), except
during the cereal monocultures of triticale (year 2012), and barley (year 2011, Fig. 7). The most negative
NBP was detected in the wheat monoculture in 2015 (NBP $\approx -108$ g C m$^{-2}$, Fig. 7), followed by the oat
and vetch mixture in 2017 (NBP $\approx -67$ g C m$^{-2}$, Fig. 7).

**4.   Discussion**
Forage species drove $CO_2$ exchange responses consistently throughout the assessed years and different
environmental conditions in the studied forage system of the Eastern Pyrenees. Cereal-legume mixtures
had more negative NEE, during the whole crop season (Fig. 3.a) and during the growth period (Fig. 3.b)
than cereal monocultures. Also, cereal-legume mixtures had lower NEE inter-annual variability
($-363$ g C m$^{-2}$, year 2013, and $-383$ g C m$^{-2}$ year 2017, Sect. 3.1) than cereal monocultures (ranging
from $-70$ to $-226$ g C m$^{-2}$, Sect. 3.1), suggesting a consistent diversity effect on NEE along different
forage mixtures and proportions of species in the mixtures.
Moreover, the diversity-interaction model (Table 2) confirmed the capacity of cereal-legume mixtures to
take up more $CO_2$, oat and vetch being the mixture with the highest net $CO_2$ uptake (Table 2). The
inclusion of legumes was key for promoting this diversity effect, since the oat and vetch mixture had a
significant effect on NEE, while the triticale addition in the mixture did not significantly increase the net
$CO_2$ uptake (Table 2).





These results agree with our first hypothesis: cereal-legume mixtures enhance the net $CO_2$ uptake in
comparison to cereal monocultures (barley, wheat and triticale). Those differences in $CO_2$ fluxes between
cereal-legume mixtures and cereal monocultures could be explained by plant species complementarity,
together with mechanisms related to ecophysiological responses, including $CO_2$ uptake and respiration
(Sect. 4.1), as well as management (Sect. 4.2).

### 342    4.1 Forage species influence on gross $CO_2$ uptake and respiration

From a mechanistic perspective, cereal-legume mixtures had higher light use efficiency than cereal
monocultures, as indicated by the slightly higher values of $GPP_{sat}$ achieved during the growth period, and
the marked $\alpha$ and $GPP_{sat}$ rebound during the fallow period (Figs. 4-5). Accordingly, cereal legume
mixtures have been reported to increase gross $CO_2$ uptake, not only via the increased photosynthesis of
legumes (Reich et al., 1997, 2003), but also increasing photosynthesis of the overall community via
nitrogen transfer from the legume to the cereal in the mixture. Interestingly, our results showed that this
increase in the gross $CO_2$ uptake and the photosynthetic activity was not accompanied by a significant
increase of daytime respiration rates ($R_{eco,day}$, Figs. 4-5).
On the other hand, $R_{eco,night}$ was clearly driven by $T_s$ and SWC (Albergel et al., 2010; Davidson and
Janssens, 2006; Yvon-Durocher et al., 2012), although it was limited at the highest $T_s$ and lowest SWC
(Fig. 6). In agreement, some authors have identified a temperature threshold at which temperature
sensitivity changes, decreasing respiration (Carey et al., 2016; Hernandez and Picon-Cochard, 2016;
Reichstein et al., 2002). This change in respiration-temperature sensitivity has been explained by
(a) changes in microbial activity (Balser and Wixon, 2009), decreasing the heterotrophic component of
$R_{eco}$; and (b) an indirect effect through limitations on GPP, resulting in limitations on the autotrophic
component of $R_{eco}$, particularly affected by the combination of high temperatures with low SWC (Niu et
al., 2012; Reichstein et al., 2002). In our study, we did not partition $R_{eco}$ into autotrophic and
heterotrophic respiration, but this shift in respiration-temperature at the highest temperatures and the
lowest SWC mostly happened after harvest (Fig. 2), which irretrievably decreased GPP and
photosynthesis, and most likely lowered the autotrophic component of $R_{eco}$ (Larsen et al., 2007).
Moreover, $R_{eco,night}$ responded similarly to $T_s$ and SWC in both cereal monocultures and cereal-legumes
mixtures, since differences in $CO_2$ respiration response to $T_s$ and/or SWC were not detected (inconsistent
differences between response parameters: $R_{eco,ref}$, $SWC_0$ , $SWC_{1/2}$ and $E_0$; see Table 3). This may well be
because although generally legumes have higher autotrophic respiration rates, with both higher leaf (Li et
al., 2016) and root respiration rates (Warembourg et al., 2003) than cereals, and there is a strong nitrogen
content – respiration relationship (Reich et al., 2008), this increase in respiration is largely driven by
higher GPP and photosynthetic activity (Larsen et al., 2007). Thus, although there had been differences in
the autotrophic respiration resulting from differences in photosynthetic rates, this does not necessarily
mean that night-time fluxes ($R_{eco,night}$) of cereal-legume mixtures had higher temperature and/or SWC
sensitivity than cereal monocultures. In addition, even if there had been differences between legume and
cereal species in their $R_{eco,night}$ sensitivity to $T_s$ and SWC, these differences were not noticeable at the
community scale (Table 3).



Interestingly, this is in line with the previously discussed NEE light response results, since the increase in
the $CO_2$ input, favoured by the presence of legumes in the community, overcompensated $CO_2$ respiration
losses, both during day ($R_{eco,day}$) and night ($R_{eco,night}$) time. This is in agreement with our second
hypothesis, cereal-legume mixtures having more negative NEE (Table 2) due to higher photosynthetic
rates, but not higher respiration rates. Chen et al. (2017) found a similar result, with legumes increasing
gross $CO_2$ uptake (higher GPP), but not enhancing $CO_2$ release, resulting in more negative NEE. Most
likely, increased total nitrogen availability, mediated by legumes, increased photosynthetic activity of the
overall community at a higher rate than respiration losses (Chen et al., 2017).

### 383 4.2 Management associated with forage types: influence on NEE and NBP

Management associated to each forage type had inherent particularities. Cereal monocultures were
harvested once the yield was sufficiently dry and grains were mature; while cereal-legume mixtures were
harvested when the vegetation was still fresh (before boot stage) for silage; the latter being a conventional
practice to improve forage nutritional value, and favour the voluntary regrowth after the harvest
(Canevari, 2000).
In our study, these differences in harvesting time resulted in clear differences in vegetation regrowth
dynamics (Fig. 2.d), which were decisive for the cumulative net $CO_2$ uptake of the whole crop season.
Thus, cereal-legume mixtures markedly regrew after the harvest, in May or early June, because the
vegetation was still in an earlier stage of phenological development, and environmental conditions were
also favourable during that time of the season. On the contrary, cereal monocultures had completed their
development cycle, and this usually left no room for voluntary regrowth after harvest (Fig. 2.d), and
hence no net $CO_2$ uptake during the fallow period (Fig. 2.a). Also, seeds that remained in the field after
the harvest did not encounter the environmental conditions required to germinate, since temperatures
were too high and SWC was too low at that time of the season, July-August.
On the other hand, all cereal-legume mixtures had a NBP that was negative during the growth period
(Fig. 7), indicating that there was C input into the system beyond the yield. In this sense, it is worth
estimating the optimum amount of biomass that can be harvested and left in the field, in order to achieve
the maximum NBP of the system, without compromising the yield. Yet, our third hypothesis had to be
rejected: cereal-legume mixtures did not clearly increase NBP as compared with cereal monocultures
during the growth period, since some cereal monocultures (wheat, year 2015, and barley, year 2016) had
a similar NBP during the growth period (Fig. 7).
However, we do still believe that cereal-legume mixtures could have shown an increase in NBP
magnitude (more negative NBP) compared with cereal monocultures, had we assessed the entire crop
season (growth and fallow). The particularly pronounced voluntary regrowth of the vegetation during the
fallow period of cereal-legume mixtures (Fig. 2.d), provided a profitable resource for livestock, besides
providing an important litter input into the system. This, combined with the moderate grazing intensity
($\approx 0.91$ LSU ha$^{-1}$), left an important part of the vegetation in the field, thereby increasing NBP, and partly
offsetting C losses due to harvesting. Thus, for future studies, we recommend to estimate C exports
through grazing during the fallow period (in addition to determine soil C content), to more accurately





estimate C inputs and exports, and consequently NBP during the whole crop season in the studied forage
system.
Finally, legumes present in cereal-legume mixtures had higher crude protein, lower neutral detergent
fibre, and higher nitrogen content than all cereals (Table S1), and vegetation remaining in the field could
also be increasing soil nitrogen. Soil nitrogen determination would also be recommendable in further
studies to fully assess the effect of forage species on soil fertility.

**Conclusions**
Based on the findings of seven years of continuous NEE measurements in an intensively managed forage
system in the Pyrenees, we found strong evidence that cereal-legume mixtures increased net $CO_2$ uptake
compared with cereal monocultures. Cereal-legume mixtures enhanced photosynthetic activity and gross
$CO_2$ uptake compared with cereal monocultures, without significantly increasing respiration, therefore
increasing net $CO_2$ uptake. Also, management practices associated with cereal-legume mixtures,
particularly an earlier harvesting time, allowed higher voluntary regrowth of the vegetation during the
fallow period. This provided additional feed for the livestock, and enhanced net $CO_2$ uptake during that
period, which was decisive for the net $CO_2$ budget of the whole crop season. Cereal-legume mixtures
enhance net $CO_2$ uptake capacity of forage systems compared with cereal monocultures, while ensuring
productivity and forage quality.

**Data availability**
Data are not public as are currently being used for other research projects. Please contact the
corresponding author by e-mail for queries concerning the data used in this study.

**Author contribution**
MI performed research, analysed data and wrote the paper; NA conceived and designed the study,
performed research and revised the paper; AR conceived and designed the study and revised the paper;
WE analysed data and revised the paper; MTS conceived and designed the study and revised the paper.
**Competing interests**
The authors declare that they have no conflict of interest.



**Acknowledgments**
We would like to thank F. Gouriveau, E. Ceschia and J. Elbers for their critical contribution to the
installation of the eddy covariance tower and to data analysis, and D. Estany and H. Sarri for field
assistance. The flux tower was installed during the FLUXPYR project (EFA34/08, INTERREG IV-A
POCTEFA, financed by EU-ERDF, Generalitat de Catalunya and Conseil Régional Midi-Pyrénées). The
following additional projects also contributed with funding to this work: CAPACITI (FP7/2007-2013
grant agreement n° 275855), AGEC 2012 (Generalitat de Catalunya), CAPAS (Spanish Science
Foundation, CGL2010-22378-C03-01), BIOGEI (Spanish Science Foundation, CGL2013-49142-C2-1-R,
supported by a FPI fellowship for Mercedes Ibañez, BES-2014-069243) and IMAGINE (Spanish Science
Foundation, CGL2017-85490-R). We would like to acknowledge the Forest Science and Technology
Centre of Catalonia (CTFC) for support with study site maintenance. We would like to thank J. Plaixats
and the Department of Animal and Food Science, Autonomous University of Barcelona for the forage
quality analyses.



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





**Tables**
**Table 1. Sward management: Forage type, species, fertilizer type (NPK 9-23-30: nitrogen 9%, phosphorus**
**23%, potassium 30%; urea; and NAC 27: calcium ammonium nitrate 27% nitrogen) and rate, sowing date**
**and rate, harvesting date, yield and C exported through yield.**

| Forage type | Species | Fertilizer (kg ha$^{-1}$) | Sowing date | Sowing rate (kg ha$^{-1}$) | Harvesting date | Yield (dry weight) | |
|---|---|---|---|---|---|---|---|
| | | | | | | (kg ha$^{-1}$) | (g C m$^{-2}$) |
| Cereal monoculture | Barley | NPK 9-23-30, 250 | 01/11/2010 | 221 | 07/07/2011 | 3000 | 138 |
| Cereal monoculture | Triticale | Urea, 140 | 01/11/2011 | 221 | 01/07/2012 | 13133 | 607 |
| Cereal-legume mixture | Triticale, oat, vetch | Not applied | 01/11/2012 | 225 | 19/06/2013 | 7500 | 339 |
| Cereal-legume mixture | Oat, vetch | Urea, 130 | 01/11/2013 | 239 | 01/07/2014 | 6720 | 304 |
| Cereal monoculture | Wheat | NPK 9-23-30, 250 Urea, 120 | 01/11/2014 | 212 | 01/08/2015 | 2580 | 118 |
| Cereal monoculture | Barley | NAC 27, 100 | 01/11/2015 | 221 | 01/09/2016 | 4500 | 208 |
| Cereal-legume mixture | Oat, vetch | Not applied | 01/11/2016 | 235 | 01/06/2017 | 7200 | 326 |







**Table 2. Diversity-interaction model results. Net ecosystem exchange (NEE) as function of air temperature**
**($T_a$), net radiation ($R_{net}$), vapour pressure deficit (VPD), and species proportions: barley, triticale, wheat, oat**
**and vetch (see forage species proportions in Fig. 1). Model performed on weekly-averaged values of all the**
**variables. Estimates (Est.) of the explanatory variables, standard error (SE), t and p-value.**

| | NEE ($\mu$mol CO$_2$ m$^{-2}$ s$^{-1}$) | | | |
| --- | --- | --- | --- | --- |
| | **Est.** | **SE** | **t** | **p** |
| **$T_a$ (ºC)** | 0.19 | 0.04 | 5.06 | < 0.001 |
| **$R_{net}$ (W m$^{-2}$)** | −0.030 | 0.002 | −12.61 | < 0.001 |
| **VPD (hPa)** | 0.17 | 0.05 | 3.56 | < 0.001 |
| **Barley (fraction)** | −1.0 | 0.3 | −3.39 | < 0.001 |
| **Triticale (fraction)** | −1.6 | 0.4 | −4.40 | < 0.001 |
| **Wheat (fraction)** | −1.5 | 0.3 | −4.42 | < 0.001 |
| **Oat x vetch (fraction)** | −2.0 | 0.3 | −7.44 | < 0.001 |
| **Triticale x oat x vetch (fraction)** | 1 | 2 | 0.58 | 0.6 |
| **$R^2_{Adj}$** | 0.45 | | | < 0.001 |







**Table 3. $R_{eco,night}$ soil temperature and soil water content response parameters based on the equations proposed**
**by Reichstein et al. (2002, Eq. 4-6): reference ecosystem respiration ($R_{eco,ref}$); soil water content below which**
**Reco ceases ($SWC_0$); soil water content at which maximal $R_{eco,night}$ halves ($SWC_{1/2}$); and a and b parameters of**
**the activation energy linear function ($E_0 = a + b \cdot SWC$). Model performed on weekly averaged values of all**
**the variables. Estimates (Est.) and standard error (SE) of the parameters. Estimates in bold are significantly**
**different from zero ($p < 0.05$).**

| Parameters | 2011 Barley | | 2012 Triticale | | 2013 Triticale, oat, vetch | | 2014 Oat, vetch | | 2015 Wheat | | 2016 Barley | | 2017 Oat, vetch | | All seasons | |
|---|---|---|---|---|---|---|---|---|---|---|---|---|---|---|---|---|
| | Est. | SE | Est. | SE | Est. | SE | Est. | SE | Est. | SE | Est. | SE | Est. | SE | Est. | SE |
| $R_{eco,ref}$ ($\mu$mol $CO_2$ m$^{-2}$ s$^{-1}$) | 1 | 2 | **4** | **2** | **2.9** | **0.3** | **2.3** | **0.2** | **2.7** | **0.6** | 9 | 15 | 3 | 2 | **2.8** | **0.3** |
| $SWC_0$ (fraction) | 0.3 | 0.6 | 0.01 | 0.02 | **0.048** | **0.005** | **0.05** | **0.002** | **0.04** | **0.03** | 0.03 | 0.06 | 0 | 0.2 | 0.01 | 0.01 |
| $SWC_{1/2}$ (fraction) | 0.4 | 0.9 | 0.1 | 0.1 | **0.054** | **0.003** | **0.052** | **0.002** | **0.08** | **0.03** | 0 | 1 | 0.1 | 0.07 | **0.06** | **0.01** |
| a (ºC$^{-1}$) | –263 | 221 | 136 | 135 | **215** | **94** | 162 | 138 | 64 | 118 | 83 | 140 | 18 | 126 | **76** | **40** |
| b (ºC$^{-1}$) | **3688** | **1645** | 596 | 1251 | –603 | 744 | 547 | 987 | **850** | **627** | -37 | 833 | 451 | 694 | **483** | **259** |
| $R^2$ | 0.59 | | 0.61 | | 0.49 | | 0.69 | | 0.75 | | 0.36 | | 0.19 | | 0.35 | |





**Figures**

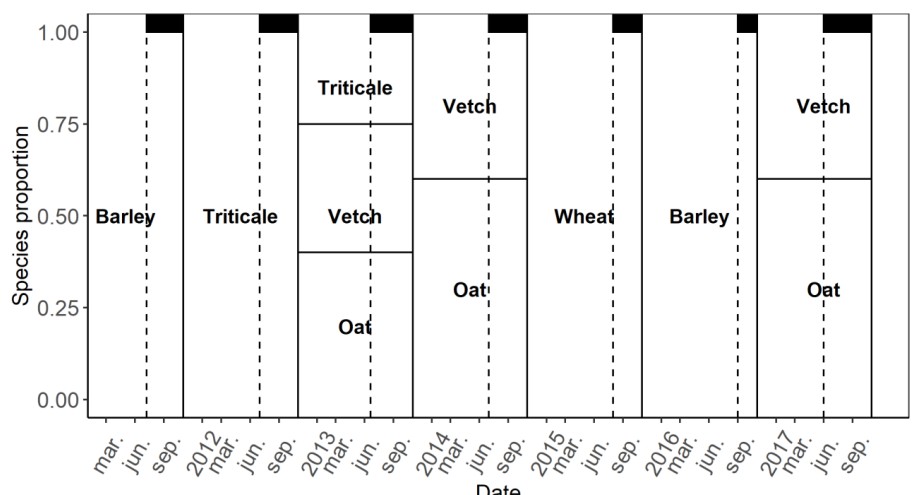

**Figure 1. Crop rotation timeline, species proportions and management events: black dashed lines indicate**
**harvesting and solid black lines indicate sowing. Top black bands indicate fallow periods in which there was**
**grazing.**


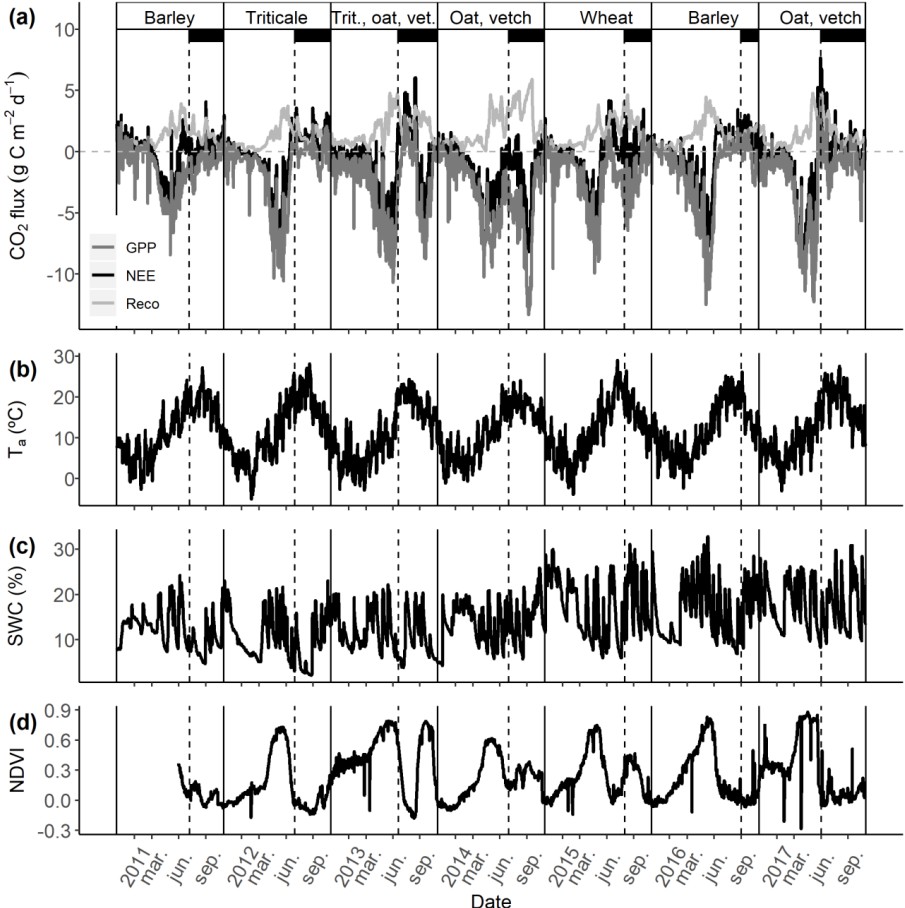

**Figure 2. Daily averaged (a) CO$_2$ fluxes: net ecosystem exchange (NEE), gross primary production (GPP) and**
**ecosystem respiration (R$_{eco}$); (b) air temperature (T$_a$); (c) volumetric soil water content (SWC); and**
**(d) normalized difference vegetation index (NDVI). Titles in the top panel indicate forage species. Black**
**dashed lines indicate harvest events and solid black lines indicate sowing events. Top black bands indicate**
**fallow periods in which there was grazing.**

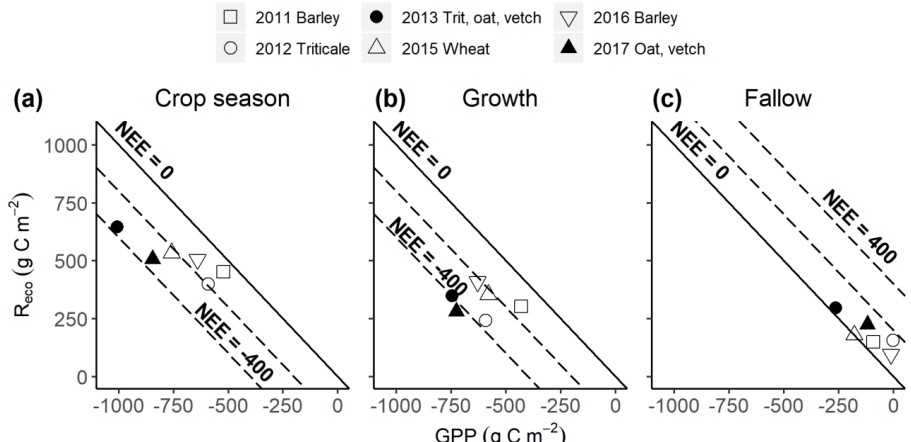

**Figure 3. Net ecosystem exchange (NEE), gross primary production (GPP) and ecosystem respiration (R$_{eco}$) budgets after gap-filling per: (a) Crop season, defined as the time from sowing to next sowing; (b) growth period, defined as the time from sowing to harvest; and (c) fallow period, defined as the time from harvest to next sowing. Solid diagonal line indicates NEE = 0 g C m$^{-2}$, dashed diagonal lines indicate ± 200 g C m$^{-2}$ NEE intervals. Open symbols indicate cereal monocultures and solid symbols cereal-legume mixtures.**





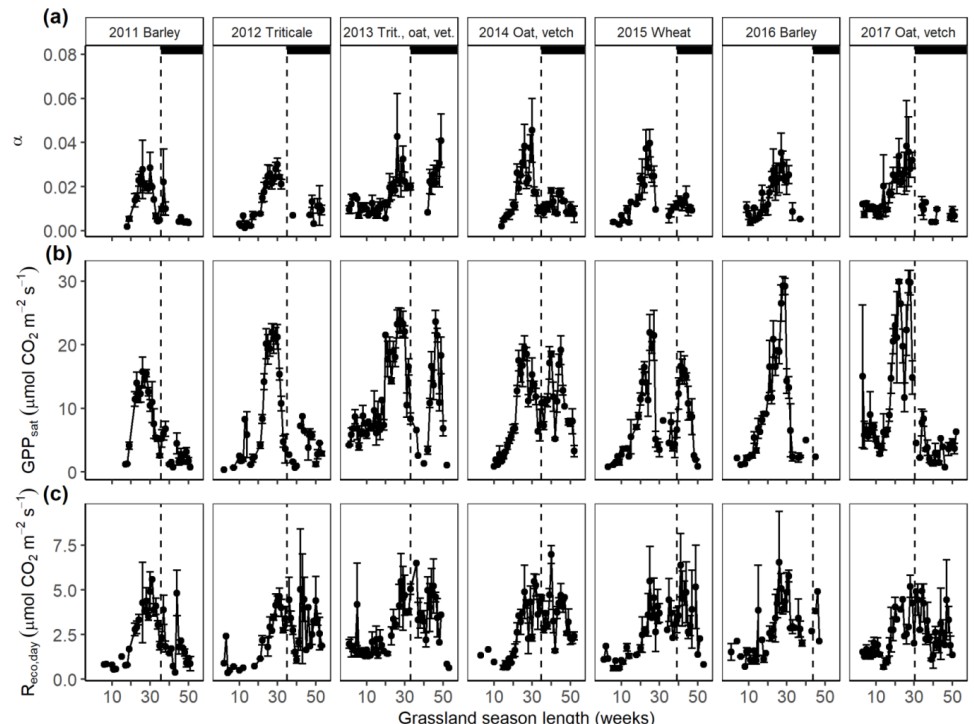

Figure 4. Seasonal dynamics of $NEE_{day}$ light response parameters Eq. (3): (a) apparent initial quantum yield ($\alpha$); (b) asymptotic gross primary production ($GPP_{sat}$); and (c) daytime ecosystem respiration ($R_{eco,day}$). Weekly averaged values and corresponding standard error bars. Titles in the top panels indicate forage species. Black dashed lines indicate harvesting events. Top black bands indicate fallow periods in which there was grazing. Gaps are due to missing data or not significant estimates ($p \geq 0.05$), which have been discarded.



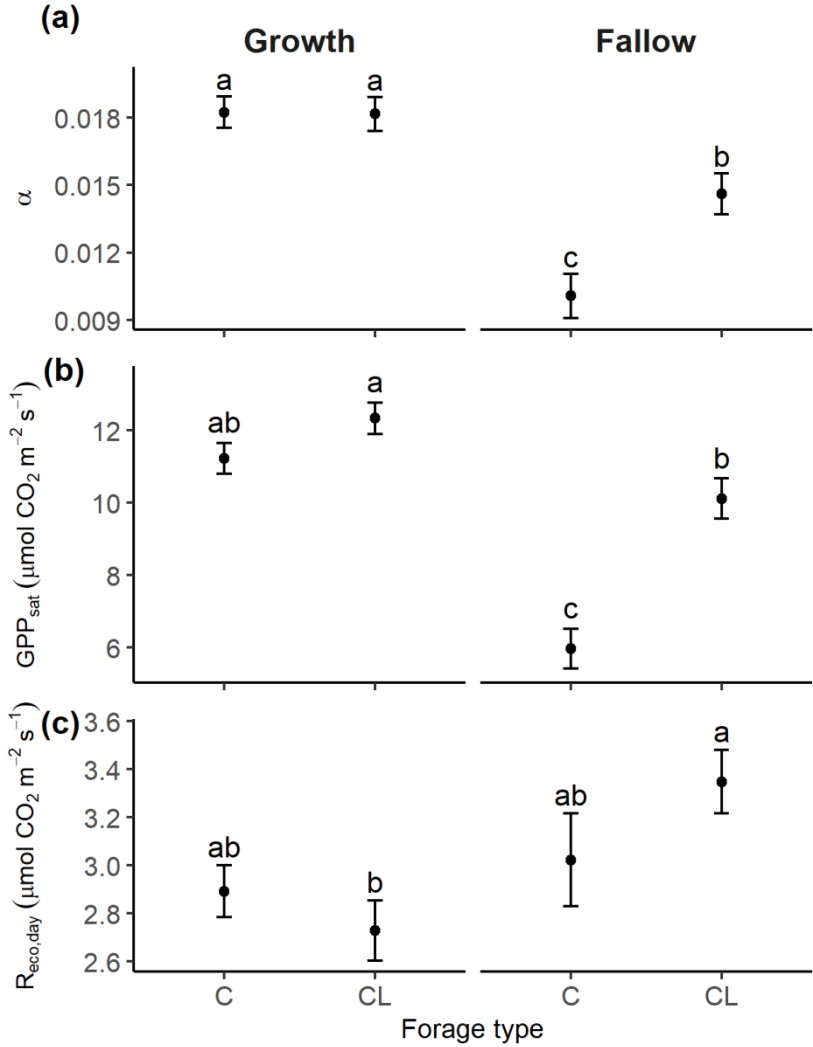

Figure 5. Light response parameters Eq. (3): (a) apparent initial quantum yield ($\alpha$); (b) asymptotic gross primary production ($GPP_{sat}$); and (c) average daytime ecosystem respiration ($R_{eco,day}$) mean ± standard error, and Tukey post-hoc test per forage type (C: cereal monoculture, CL: cereal-legume mixture) and period (growth and fallow). Letters indicate significant differences among groups ($p < 0.05$). See ANOVAs results in Table S3.

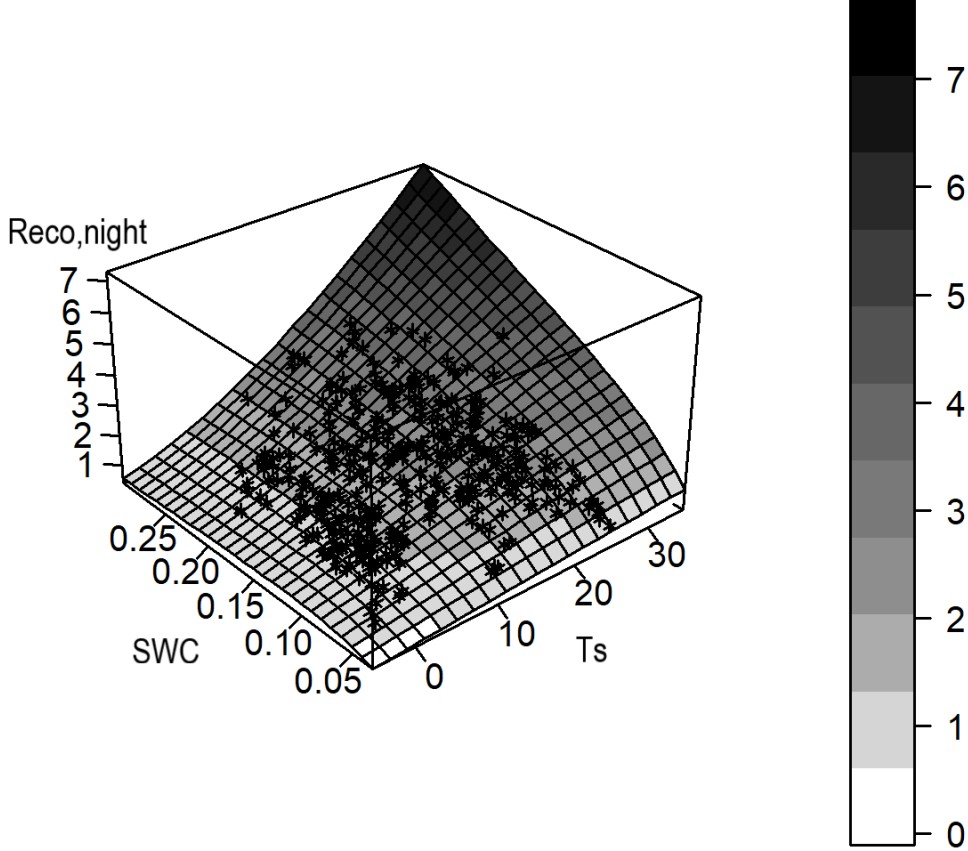

**Figure 6. R$_{eco,night}$ trend surface as a function of soil temperature (T$_s$) and soil water content (SWC), by the**
**equations proposed by Reichstein et al. (2002, Eq. 4-6). Model performed on weekly averaged data of all the**
**variables. The grid shows the trend surface and dots are observed data.**

Biogeosciences Open Access
Discussions
EGU

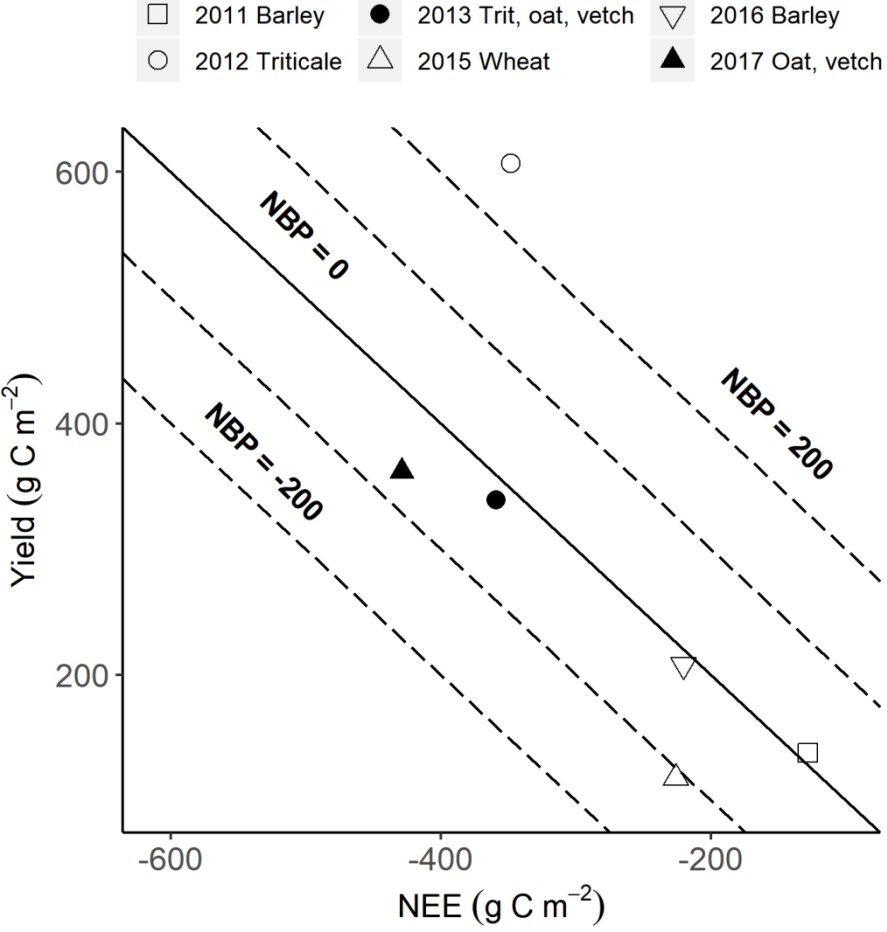

**Figure 7. Net biome production (NBP), net ecosystem exchange (NEE) and yield during the growth period, defined as the time from sowing to harvest. Solid diagonal line indicates NBP = 0 g C m$^{-2}$, dashed diagonal lines indicate ± 100 g C m$^{-2}$ NBP intervals. Open symbols indicate cereal monocultures and solid symbols cereal-legume mixtures.**