# Peer review of "Cereal-legume mixtures increase net CO$_2$ uptake in a forage system of the Eastern Pyrenees"

_Biogeosciences, 2020_

## Referee Comment (RC1) · Anonymous Referee #1 · 24 Jul 2020

General comments on work of Ibanez et al This article presents the CO2 exchanges between vegetation cover and the atmosphere as a function of their diversity and environmental variables over 6 years of study. The choice of vegetation cover and its management with herbivores was reasoned in a very relevant way in relation to agricultural practices and agro-ecological issues. The results clearly show the positive effect of maintaining plant cover throughout the year on the fixation of C and its storage into the ecosystem. The authors analyse the mechanisms of C fixation in agrosystems in some detail. Therefore, the findings of this manuscript are important, timely and of interest of BG readers. Nevertheless, the manuscript in its present form suffers from many limitations that need to be carefully considered before any decision can be made regarding its publication. There are serious shortcomings in the description of studied

sites and methods. There is no description of the treatments of the agricultural practices, the number of replicates per treatment, the size of the plots and more generally of the experimental design. The status of studied sites are unclear. Are the studied sites experimental sites managed by the research team or fields from farmers that the research used in their experiment? Concerning the measurement of $CO_2$ exchanges, the existence of a flow tower is mentioned, but we do not understand how one sole flow tower will make it possible to monitor the $CO_2$ exchanges induced by different plant covers.

The description of the yield estimation method is imprecise, even though this variable is essential for estimating the agronomic relevance and carbon balance of the studied plant covers. Authors wrote that yield was estimated (Table 1) considering the productivity reported by the manager and in situ samplings after oven drying plant material at 60 °C until constant weight. How did the manager proceed to estimate the yield? How was the forage production estimated in the presence of grazers? How did you manage the space-and-time variability in yield (Frequency of measurements etc)? Section 2.1 of the M&M section.

Some findings seem to not be supported by the data, or the data are not enough clearly presented. In the summary the authors write "Overall, cereal-legume mixtures enhanced net $CO_2$ sink capacity of the forage system, while ensuring productivity and forage quality" but when we read the description data L320-322 on the ecosystem C storage "The most negative NBP was detected in the wheat monoculture in 2015 (NBP ≈ −108 g C m−2321 , Fig. 7), followed by the oat and vetch mixture in 2017 (NBP ≈ −67 g C m−2322 , Fig. 7)." the key role of legumes does not appear.

L42 Its seems to me that the word "voluntary" can be removed L46 What do you mean exactly with "beside the yield"? We do not see the link with the main idea of the sentence that these systems store C. Maybe you would like to say you accounted the C exported by harvests in the final C balance that indicated that these systems fix C in biomass and soils? Need to be clarified. L86-89 It is an interesting approach to suggest working hypotheses at the end of the introduction. However, these hypotheses are given without explanations on the expected cause-effect link and thus have little scientific value. Please better introduce these hypotheses. L165 Could you define SWC again to help readers in the forest of acronyms. L176-182 Based on the absence of SWC effect in the null model, did you also removed the SWC from the diversity model before analysis? If yes I am not it is correct because the effect of SWC might appear when the diversity of plants are accounted. More generally, don't you find strange this lack of effect of SWC although the availability of water seems very limiting for ecosystem functioning in Mediterranean basin? L330-331 You should be more specific on the effect. The diversity stabilized the NEE over the environmental fluctuations supporting the insurance-hypothesis of biodiversity. L365-369 This sentence is too long and contains too many ideas to be understood. Please split it in several distinct sentences and clarify ideas. L377-379 Concerning your statement "This is in agreement with our second hypothesis, cereal-legume mixtures having more negative NEE (Table 2) due to higher photosynthetic rates, but not higher respiration rates.". In fact it is not so obvious. If I am not wring, you have one mixture with legumes (the last one of the Tables) inducing positive NEE, that is, losing carbon. Therefore, I am not completely convinced by your statement for the moment. L389 and at many other places it seems that your text is shifted on the line, is there a problem with text formatting? L398-404 I find this part to follow. . .I do not see well the logical link between the beginning and the end of the paragraph. What is the main message here? L405-414 A similar comment here. We do not well understand what have been measured/observed, what are speculations. Could you better separate the different ideas and identify the list of things to do to better assess the carbon balance of these various agricultural practices.

---

## Referee Comment (RC2) · Anonymous Referee #2 · 25 Aug 2020

Investigations aimed at improving C-sequestration in soil through adapted land use are of great importance because they could make an important contribution to the short- and medium-term mitigation of anthropogenic climate change. At present there are still too many knowledge deficits to fully exploit the potential of this approach. In view of this, the authors' intention to contribute to the solution of this problem is very reasonable and logical (introduction, lines 52-77). This also applies to the selection of forage cultivation systems and the use of the eddy-covariance technique for conducting the investigations. Unfortunately, however, the manuscript is characterized by two serious deficiencies. These are so fundamental and at the same time irreparable that publication of the study in Biogesciences cannot be recommended.

The chosen experimental approach does not reflect the current state of the art. This is

comprehensively described and discussed e.g. in the publications of Smith et al. 2010, Agriculture, Ecosystems & Environment, 139, 302–315; Soussana et al. 2010, Animal, 4, 334–350; Chenu et al. 2018, Soil Tillage Res. 188, 41–52; Smith et al. 2020, Global Change Biology, 26, 219–241. If the authors had followed this approach, they would never have come up with the idea to characterize the climate impact of crop species solely on the basis of several months of NEE fluxes and seasonal NBP budgets. In order to determine the influence of crop species on the context-relevant C sequestration (longer-term storage of $CO_2$-C in the soil's C stock), annual NBP budgets would have had to be determined over a period of several years. Only then is it possible to avoid bias of the results due to the different temporal dynamics of plant C-input and C-output via soil C mineralization and interanual weathering variability. To be on the safe side, the $CO_2$ flux-based approach is now also combined with direct measurements of changes in the soil C stock. Since this did not happen, the authors have missed the self-set goal of their investigations. This is also indirectly admitted at the end of the discussion (lines 405-414).

Contrary to the authors' assertions, the experimental approach used is only suitable to a very limited extent for clarifying the question of whether grain-legume mixtures represent a stronger $CO_2$ sink than grain monocultures. Clear statements on this would have required the simultaneous investigation of cereal monocultures and cereal-legume mixtures. Since the authors have only examined the different cultivation variants one after the other in a crop rotation, they are not able to separate the direct effect of the respective crop on the $CO_2$ source function from the indirect preceding crop effect and the influence of the current annual weather. In addition, the form and amount of fertilizer applied varied between years, even with the same crop. This is a clear violation of the ceteris paribus principle, one of the most important prerequisites for obtaining clear results in experimental research. With the help of the diversity interaction model used, it is only partially possible to compensate for this deficit. This is because, when determining the so-called species-specific effects, the effect of the current random variation of the other factors is inevitably included. Finally, only the expected but trivial statement remains that the prolonged presence of photosynthetically active plants during the vegetation period can lead to a temporary improvement of the $CO_2$ sink function.

Minor deficits

A lot of information and data that are important for the interpretation of the results are missing. This is especially true for

- Description of the study site: physical and chemical properties of the entire soil profile, cultivation history

- Type and timing of tillage

- Dealing with the above-ground phytomass: what is behind the yield and the C export? Only the amount of grain harvested or always the total above-ground phytomass? What and how much remained on the field in the form of harvest residues?

In the case of triticale, results do not seem to be consistent. It is not plausible that GPP (Figure 3b) should be somewhat lower than the C yield (Table 1).

Please also note the supplement to this comment:
https://bg.copernicus.org/preprints/bg-2020-173/bg-2020-173-RC2-supplement.pdf

---

## Author Comment (AC1) · 2 Oct 2020

**Reviewer's comments in blue**

**Authors' answers in black**

**Reviewer 1**

There is no description of the treatments of the agricultural practices, the number of replicates per treatment, the size of the plots and more generally of the experimental design. The status of studied sites are unclear. Are the studied sites experimental sites managed by the research team or fields from farmers that the research used in their experiment?

Table 1 shows main agricultural practices, including sown species, type and amount of fertilizer, sowing dates and rates, harvesting dates and yield. There are no simultaneous plots because this is a crop field, managed by farmers, and equipped with an eddy covariance station placed in the middle of the field (see pictures). Every year the field is sown and harvested with the corresponding species (Figure 1). The size of the plot is the whole field (21.70 ha), although the eddy covariance station is obviously recording $CO_2$ exchange according to the corresponding footprint, which can be considered representative for the entire field. In the revised paper we will specify this point more clearly.

[Figure]

[Figure]

Concerning the measurement of CO₂ exchanges, the existence of a flow tower is mentioned, but we do not understand how one sole flow tower will make it possible to monitor the CO₂ exchanges induced by different plant covers.

There are not simultaneous different plant covers. Plant covers change every year, see Table 1 for sowing and harvesting dates, and Figure 1, which is a scheme of the crop rotation timeline.

The description of the yield estimation method is imprecise, even though this variable is essential for estimating the agronomic relevance and carbon balance of the studied plant covers. Authors wrote that yield was estimated (Table 1) considering the productivity reported by the manager and *in situ* samplings after oven drying plant material at 60 ∘C until constant weight. How did the manager proceed to estimate the yield?

The manager knows exactly the productivity, because the whole yield is harvested and processed in straw bales for production purposes. This is a commercial farming field. We dried a subsample of the yield at 60°C until constant weight to determine the remaining water content in the yield reported by the manager, which we could then convert to dry matter yield. In this way we standardized the productivity between years. This information will be added in the revisions.

How was the forage production estimated in the presence of grazers?

Grazers were only present during the fallow period, after the harvest. However, the forage production during that period and under the presence of grazers could not be estimated satisfactorily. That is why we are presenting the net biome production (NBP) only during the growth period (from sowing to harvesting).

Yet, we consider our net ecosystem exchange (NEE) results during the fallow period (presence of grazers) accurate. The eddy covariance station measures the net $CO_2$ uptake within the footprint and considering the size of the plot and the low grazing pressure the probability of finding grazer animals inside the footprint is low. However, if a grazer was within the footprint (which is normally not the case) then the net production can be slightly underestimated, whereas if the grazers are not in the footprint then the measurements should very well represent productivity. We can discuss this in more detail in the revision to clarify.

How did you manage the space-and-time variability in yield (Frequency of measurements etc)? Section 2.1 of the M&M section.

In terms of yield there is no much space variability, since the field is all sown and harvested at the same time. Therefore, the yield is the productivity in the period from sowing to harvesting. The temporal variability over that time is however well resolved with the continuous eddy covariance flux measurements and is fully considered in the study.

Some findings seem to not be supported by the data, or the data are not enough clearly presented. In the summary the authors write "Overall, cereal-legume mixtures enhanced net $CO_2$ sink capacity of the forage system, while ensuring productivity and forage quality" but when we read the description data L320-322 on the ecosystem C storage "The most negative NBP was detected in the wheat monoculture in 2015 (NBP ≈ −108 g C m−2321 , Fig. 7), followed by the oat and vetch mixture in 2017 (NBP ≈ −67 g C m−2322 , Fig. 7)." the key role of legumes does not appear.

In terms of $CO_2$ up-take, the $CO_2$ sink capacity was clearly enhanced with the presence of legumes, as supported by our net ecosystem exchange (NEE) results, summarized in Figure 2, and the significant oat x vetch interaction in Table 2. Those agree with the sentence "Overall, cereal-legume mixtures enhanced net $CO_2$ sink capacity of the forage system, while ensuring productivity and forage quality".

On the other hand, when we talk about net biome production (NBP, C uptake into the system) certainly there were not clear differences between cereal monocultures and cereal-legume mixtures. As we state in the discussion lines 401- 404: "Yet, our third hypothesis had to be rejected: cereal-legume mixtures did not clearly increase NBP as compared with cereal monocultures during the growth period, since some cereal monocultures (wheat, year 2015, and barley, year 2016) had a similar NBP during the growth period (Fig.7). However, as we also state in the lines 405-4011. "we do still believe that cereal-legume mixtures could have shown an increase in NBP magnitude (more negative NBP) compared with cereal monocultures, had we assessed the entire crop season (growth and fallow). The particularly pronounced voluntary regrowth of the vegetation during the fallow period of cereal-legume mixtures (Fig.2.d), provided a profitable resource for livestock, besides providing an important litter input into the system. This, combined with the moderate grazing intensity ($\approx$ 0.91 LSU ha$^{-1}$), left an important part of the vegetation in the field, thereby increasing NBP, and partly offsetting C losses due to harvesting.

L42 It seems to me that the word "voluntary" can be removed

The term "voluntary" refers to plants that grow on their own, rather than being deliberately planted by a farmer. Volunteers often grow from seeds that remain in the field, and unlike weeds — which are unwanted plants — a volunteer may be encouraged by farmers once it appears (Davey, J., 2007).

*Davey, J. (2007). "Crop Ferality and Volunteerism". Annals of Botany. 99: 205–206. doi:10.1093/aob/mcl244. PMC 2802985.*

However, the word can be easily removed if needed. Alternatively, we suggest to add a short explanation with the Davey (2007) reference to clarify this term.

L46 What do you mean exactly with "beside the yield"? We do not see the link with the main idea of the sentence that these systems store C. Maybe you would like to say you accounted the C exported by harvests in the final C balance that indicated that these systems fix C in biomass and soils? Need to be clarified.

Yes, we mean that in addition to the C exported through the yield itself, the field stores C in the remaining biomass and/or the soil. We will rephrase the sentence according to the reviewer's suggestion.

L86-89 It is an interesting approach to suggest working hypotheses at the end of the introduction. However, these hypotheses are given without explanations on the expected cause-effect link and thus have little scientific value. Please better introduce these hypotheses.

We hypothesize that: :

(1) cereal-legume mixtures will show higher net $CO_2$ uptake (more negative NEE) in comparison to cereal monocultures

Forage mixtures have been generally associated with higher productivity than monocultures (Brophy et al., 2017; Finn et al., 2013; Kirwan et al., 2007; Ribas et al., 2015), resulting from higher resource use efficiency, including light (Hofer et al., 2017; Milcu et al., 2014), water (Chapagain and Riseman, 2015; Liu et al., 2016), and nitrogen (Sturludóttir et al., 2013; Suter et al., 2015). Considering these facts, we presume a link between mixtures and a higher $CO_2$ uptake (more negative NEE) compared to monocultures.

(2) the increase in net $CO_2$ uptake in mixtures compared to monocultures will be related to increased GPP in combination with unchanged $R_{eco.}$

Legumes present higher light use efficiency than cereals (Hofer et al., 2017; Milcu et al., 2014), and at the same time they stimulate the photosynthetic rates of the overall community. Since, there is an increase in the productivity linked to the presence of legumes in the mixture the gross uptake favoured by the higher light use efficiency must exceed respiration losses.

And,

(3) mixtures will show more negative NBP compared to monocultures.

We hypothesize that the productivity enhancement results in an increased belowground biomass production, as well as more plant residues that are not harvested, which can be both incorporated into the soil, increasing the C input into the system.

We can explain all three hypotheses in a more precise and explicit way in the revised version.

L165 Could you define SWC again to help readers in the forest of acronyms.

Yes, soil water content (SWC) can be spelled out again.

L176-182 Based on the absence of SWC effect in the null model, did you also removed the SWC from the diversity model before analysis? If yes I am not it is correct because the effect of SWC

might appear when the diversity of plants are accounted. More generally, don't you find strange this lack of effect of SWC although the availability of water seems very limiting for ecosystem functioning in Mediterranean basin?

Yes, the effect of SWC was tested also when the diversity terms were included, and still it did not have a significant effect. The most parsimonious and explicative model included then air temperature ($T_a$), net radiation ($R_{Net}$), vapour pressure deficit (VPD) and composition terms (proportion of the given species, Equation 2). The lack of a SWC effect is probably related to the variability explained by the VPD. This suggests the water availability effect was better explained by the VPD than by the SWC in our dataset.

L330-331 You should be more specific on the effect. The diversity stabilized the NEE over the environmental fluctuations supporting the insurance-hypothesis of biodiversity.

Yes, thanks for the comment, this is correct; we will add more specific text to clarify this aspect.

L365-369 This sentence is too long and contains too many ideas to be understood. Please split it in several distinct sentences and clarify ideas.

We agree and we will simplify the sentence and clarify the ideas. We suggest the following revised version:

However, this increase in respiration is largely driven by higher GPP and photosynthetic activity (Larsen et al., 2007), which results in higher autotrophic respiration rates in mixtures than those in cereals.

L377-379 Concerning your statement "This is in agreement with our second hypothesis, cereal-legume mixtures having more negative NEE (Table 2) due to higher photosynthetic rates, but not higher respiration rates.". In fact it is not so obvious. If I am not wrong, you have one mixture with legumes (the last one of the Tables) inducing positive NEE, that is, losing carbon. Therefore, I am not completely convinced by your statement for the moment.

The interaction oat x vetch x triticale (Table 2) is not significant, then although the estimate is positive, we cannot assume that this combination of species is losing carbon. In fact, this means that the interaction between oat and vetch is explicative enough, and we are not adding explained variability through the addition of tricticale in the mixture. On the other hand, we did

detect some differences in the NEE light response ($\alpha$, $GPP_{sat}$), specially during the fallow period, while $R_{eco,day}$ did not increase (Figure 5). At the same time, we did not detect consistent differences in the $R_{eco,night}$ parameters between cereal monocultures and cereal-legume mixtures (Table 3). According to these results we can infer higher photosynthetic rates and higher gross uptake, but not higher respiration rates, supporting our second hypothesis. However, we can see that we have to clarify the text in this aspect and we will do so.

L389 and at many other places it seems that your text is shifted on the line, is there a problem with text formatting?

We can easily solve this.

L398-404 I find this part to follow. . .I do not see well the logical link between the beginning and the end of the paragraph. What is the main message here?

We suggest the following revised version:

On the other hand, all cereal-legume mixtures had a NBP that was negative during the growth period indicating that there was C input into the system beyond the yield. However, our third hypothesis had to be rejected: cereal-legume mixtures did not clearly increase NBP as compared to cereal monocultures during the growth period, since some cereal monocultures (wheat, year 2015, and barley, year 2016) had a similar NBP during the growth period (Fig. 7).

L405-414 A similar comment here. We do not well understand what have been measured/observed, what are speculations. Could you better separate the different ideas and identify the list of things to do to better assess the carbon balance of these various agricultural practices.

We agree and we suggest the following revised version:

Even so, we do still believe that cereal-legume mixtures could have shown an increase in the NBP (more negative NBP) compared with cereal monocultures, had we assessed the entire crop season (growth plus fallow). In agreement to this hypothesis, our NDVI results showed that cereal-legume mixtures performed a particularly pronounced voluntary regrowth of the vegetation during the fallow period (Fig. 2.d), which provided an important litter input into the system. Thus, we infer that the vegetation remaining in the field probably increased the C input

into the soil, partly offsetting ecosystem C losses due to harvesting. For future studies, we recommend to estimate C exports through grazing during the fallow period (in addition to determine soil C content), to more accurately estimate C inputs and exports, and consequently NBP during the whole crop season in the studied forage system.

---

## Author Comment (AC2) · 2 Oct 2020

**Reviewer's comments in blue**

**Authors' answers in black**

**Reviewer 2**

Unfortunately, however, the manuscript is characterized by two serious deficiencies. These are so fundamental and at the same time irreparable that publication of the study in Biogesciences cannot be recommended.

(1) The chosen experimental approach does not reflect the current state of the art. This is comprehensively described and discussed e.g. in the publications of Smith et al. 2010, Agriculture, Ecosystems & Environment, 139, 302–315; Soussana et al. 2010, Animal, 4, 334–350; Chenu et al. 2018, Soil Tillage Res. 188, 41–52; Smith et al. 2020, Global Change Biology, 26, 219–241. If the authors had followed this approach, they would never have come up with the idea to characterize the climate impact of crop species solely on the basis of several months of NEE fluxes and seasonal NBP budgets. In order to determine the influence of crop species on the context-relevant C sequestration (longer-term storage of CO2-C in the soil's C stock), annual NBP budgets would have had to be determined over a period of several years. Only then is it possible to avoid bias of the results due to the different temporal dynamics of plant C-$_{input}$ and C$_{output}$ via soil C mineralization and interanual weathering variability. To be on the safe side, the CO2 flux-based approach is now also combined with direct measurements of changes in the soil C stock. Since this did not happen, the authors have missed the self-set goal of their investigations. This is also indirectly admitted at the end of the discussion (lines 405-414).

As we state at the end of our introduction our main goals were: "To assess differences between cereals grown in monoculture and cereal-legume mixtures in (1) ecosystem-scale $CO_2$ fluxes, for the whole crop season and separately for the two periods of growth and fallow; (2) potential sensitivities of $CO_2$ exchange related to short-term variations in light, temperature and soil water content; and (3) NBP during the growth period. Also, we hypothesize that cereal-legume mixtures in comparison to cereal monocultures: (1) will show more net $CO_2$ uptake (more negative NEE); (2) this increase in the net uptake will be due to increased GPP in combination with unchanged $R_{eco}$; and (3) will show more negative NBP.

Thus, assessing differences on the net biome production (NBP) was certainly one of our goals, but not the central one. We agree that to fully assess the NBP, soil C content determination is needed. However, we could do an estimation of the NBP during the growth period, according to the available data, and this is openly explained in the methods and afterwards discussed. Yet, our NBP estimation is providing an extra information on how the different crops perform in terms of C balance. This approach is useful in terms of comparison among management treatments, and we are cautious and fair in setting the boundaries within which it has to be interpreted. Moreover, our analysis on the net ecosystem exchange (NEE) is providing valuable information for improving forage production and management in the Mediterranean context. Indeed, in agreement with this reviewer, investigations aiming at improving C-sequestration in soil through adapted land use are of great importance because they could make an important contribution to the short- and medium-term mitigation of anthropogenic climate change. Our suggestion is then to modify the focus of our objectives and hypothesis to avoid wrong expectations, and to clarify in the discussion which components are still missing to obtain the integrative insight that this reviewer expected.

(2) Contrary to the authors' assertions, the experimental approach used is only suitable to a very limited extent for clarifying the question of whether grain-legume mixtures represent a stronger CO2 sink than grain monocultures. Clear statements on this would have required the simultaneous investigation of cereal monocultures and cereal-legume mixtures. Since the authors have only examined the different cultivation variants one after the other in a crop rotation, they are not able to separate the direct effect of the respective crop on the CO2 source function from the indirect preceding crop effect and the influence of the current annual weather. In addition, the form and amount of fertilizer applied varied between years, even with the same crop. This is a clear violation of the ceteris paribus principle, one of the most important prerequisites for obtaining clear results in experimental research. With the help of the diversity interaction model used, it is only partially possible to compensate for this deficit. This is because, when determining the so-called species-specific effects, the effect of the current random variation of the other factors is inevitably included. Finally, only the expected but trivial statement remains that the prolonged presence of photosynthetically active plants during the vegetation period can lead to a temporary improvement of the $CO_2$ sink function.

Our study area is a crop field, used for forage production, and we aimed to assess what was happening under the typical circumstances of a crop rotation system of these characteristics. Certainly, this is a "survey" type experimental site, and our study was observational, rather than a "manipulative experiment" (trials) which are another concept. Both have their strengths and weaknesses. The benefit from survey type experiments is that they tend to be closer to real-world conditions since the system "as is" is explored. We see that we have to clarify this in more detail to avoid wrong expectations, and we have to clarify early on what such a study can potentially contribute to the understanding of the system, and what it cannot.

However, we want to emphasize that to overcome the interannual weathering and management (as for instance fertilizer application) variability, we have used different approaches to assess the forage type effect on our variables of interests. First, we have performed $CO_2$ budgets (annual, crop and fallow period). Second, we have used a diversity interaction modelling approach, in which we have included environmental variables, such as air temperature, vapor pressure deficit (VPD), etc. considering such interannual weathering variability. Also, tested the random interannual variability in this modelling including the variable "time", which was not significant, and thus removed from the final model. And third, we have run a mechanistic approach modelling for the light response and the ecosystem respiration response to temperature and soil water content. These complementary methodologies have shown consistent results, cereal-legume mixtures performing a higher $CO_2$ net uptake than cereal monocultures, in spite of the interannual variability.

**Minor deficits**

These will be addressed in the revised version.